# Anti-Kasha triplet energy transfer and excitation wavelength dependent persistent luminescence from host-guest doping systems

Weiwei Xie[1,4], Wenbin Huang[2,4], Jietai Li[1], Zikai He [2] ✉, Guangxi Huang [1] ✉, Bing Shi Li [1] ✉ & Ben Zhong Tang [3] ✉

Anti-Kasha's process in organic luminogens has attracted many attentions since its discovery. However, only limited examples of anti-Kasha's rule have been reported and anti-Kasha triplet energy transfer (ET) is even less-touched. Benefiting from anti-Kasha's rule, this work provided an efficient strategy to realize excitation wavelength dependent (Ex-De) afterglow in a host-guest system. The **host** has almost imperceptible RTP upon 365 nm excitation and **guest** is totally RTP inactive, while the doping host-guest system exhibits Ex-De afterglow with improved quantum yields. Anti-Kasha triplet ET process is realized from the higher excited triplet state $T_2$ of **host** to the lowest excited singlet state $S_1$ of the aggregated/unimolecular **guest**. ET efficiency in the doping system could be tuned by simply changing its processing methods to guide **host** and **guest** to adopt denser or looser intermolecular packing. The strategy of anti-Kasha triplet ET endows the host-guest doping system with multiple stimuli-responsive properties, including Ex-De afterglow, mechano-, and thermal-triggered afterglow behaviors. The corresponding applications of these properties are also realized in multiple information anti-counterfeiting and display.

Organic room temperature phosphorescence (RTP) systems with long lifetime afterglow have been deeply investigated and widely applied in various areas, including organic optoelectronics, chemical biomedicine, optical sensing, and information encryption[1–3]. Excellent phosphorescence has been realized using various strategies through boosting intersystem crossing (ISC) from excited singlet state ($S_1$) to triplet states ($T_n$) and meanwhile inhibiting nonradiative decays, including heavy atom effect[4–6], El-Sayed's rule[7,8], crystallization engineering[9–13], recombination of charge-separated states[14–17], energy transfer (ET)[18,19], hydrogen-bonding interaction[20–23], through-space conjugation[24–26], polymer matrix assistance[27–33], resonance activation[34,35], multicomponent combination[36–42], and etc. The emission behaviors of organic RTP materials are sensitive to temperature, oxygen, light, or humidity, so that they can be regulated by external stimuli, and have versatile potential applications. However, RTP systems with multiple stimuli-responsive behaviors are still less developed, and their single function could not meet complicated practical needs[43].

[1]Key Laboratory of New Lithium-Ion Battery and Mesoporous Material, College of Chemistry and Environmental Engineering, Shenzhen University, 1066 Xueyuan Avenue, Nanshan District, Shenzhen, Guangdong, China. [2]School of Science, Harbin Institute of Technology, Shenzhen, HIT Campus of University Town, Shenzhen, China. [3]School of Science and Engineering, Shenzhen Institute of Aggregate Science and Technology, The Chinese University of Hong Kong, Shenzhen, Guangdong, China. [4]These authors contributed equally: Weiwei Xie, Wenbin Huang. ✉e-mail: hezikai@hit.edu.cn; huanggx@iccas.ac.cn; phbingsl@szu.edu.cn; tangbenz@cuhk.edu.cn

Tunable RTP emission, particularly excitation wavelength dependent (Ex-De) afterglow, provides an opportunity to explore multiple stimuli-responsive materials with potential prospects in optical sensing[44–49]. Ex-De luminescence is mainly attributed to the emission from different photophysical processes that requires controlled expressions of excitation dynamics. At present, it is mainly divided into three categories: emission from different excited states in single-component systems[50–56], emission from different components in multicomponent systems[57–62], and emission from different aggregated states in cluster systems[63–74]. The most practical strategy to achieve tunable Ex-De luminescence is utilizing multicomponent systems with the advantages in facile preparation, conveniently tunable process and wide applicability.

Among small molecular multicomponent systems, Ex-De afterglow is less explored. In 2021, Huang's group reported several ultralong organic host/guest phosphorescence materials with dynamic lifetime-tuning properties, which was obtained by introducing different hosts with tuning triplet energy levels in order to boost ISC and ET processes, and Ex-De phosphorescence was demonstrated[61]. However, considerable efforts usually focus on ET of the lowest excited state but seldom on higher excited state. If the utilization of higher excited state energy could be realized, Ex-De afterglow based on unusual anti-Kasha's rule would be feasible in multicomponent systems. Anti-Kasha's rule suggests that photophysical and photochemical processes in condensed phase can be carried out in higher excited state of a given multiplicity[75,76]. However, most excitons are unstable in higher excited state and they easily transfer to the lowest excited state through internal conversion (IC), therefore, it is difficult to detect the process of anti-Kasha's rule. Up to now, only limited examples of anti-Kasha's rule have been reported[77,78] and ET in higher excited state is even less-touched[79,80]. The higher triplet states are more susceptible to environmental quenching and non-radiative interference, hence detectable anti-Kasha emission is only limited in a low-temperature and inert environment. If grievous quenching of higher excited state excitions could be overcome and multiple radiative processes in anti-Kasha system were reasonably utilized, it will be feasible to achieve ET from higher triplet state $T_2$ to construct new Ex-De afterglow systems.

Herein, this work provided an efficient way to realize Ex-De afterglow by ET process in a host-guest system using a simple doping strategy. The ET process is accomplished from the higher excited triplet state $T_2$ of the **host** to the lowest excited singlet state $S_1$ of the aggregated/unimolecular **guest** (Fig. 1a). The **host** has almost imperceptible RTP upon 365 nm excitation and the **guest** is totally RTP inactive (Fig. 1b), while the doping host-guest system exhibits Ex-De afterglow with improved quantum yields (Figs. 1b and S1). ET efficiency in the doping system could be tuned by changing intermolecular packing degree depending on different doping treatments. The strategy of anti-Kasha triplet ET endows the doping system with multiple stimuli-responsive properties, including Ex-De afterglow, mechano-, and thermal-triggered afterglow behaviors. Based on these features, corresponding applications of these properties are also explored in multiple information anti-counterfeiting and display.

## Results
### Preparations of host, guest, and doping systems
Both the commercially available 9(10H)-acridone and diphenylsulfone were purified by two-step recrystallization process with different solvent conditions. The chemical structure of two compounds was

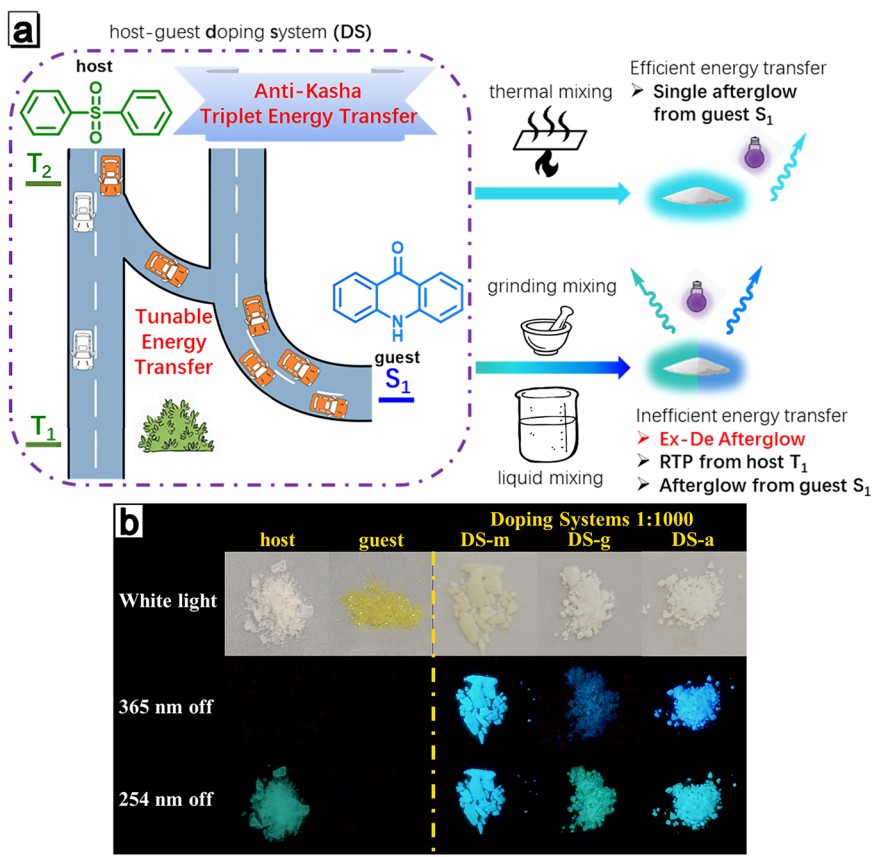

**Fig. 1 | Host-guest doping systems with Ex-De afterglow through anti-Kasha triplet energy transfer. a** The high-energy excitons (represented by 'car') from $T_2$ state of **host** can transfer to $S_1$ state of **guest** (represented by 'side road') through anti-Kasha triplet ET or to its own $T_1$ state (represented by 'main road') through IC. The transition between inefficient and efficient ET process can be achieved by employing distinct processing methods, resulting in multiple variation of emission behaviors. **b** Images of **host**, **guest**, **DS-a**, **DS-g** and **DS-m** with doping ratio of 1:1000 before and after 365/254 nm UV irradiation under ambient conditions.

verified with NMR (Figs. S23–S26) and their high purity was proved by high performance liquid chromatography (HPLC) (Figs. S2 and S3). Thermogravimetric analysis of diphenylsulfone and 9(10H)-acridone revealed their high thermal stability with the decomposition temperature more than 200 and 300 °C (Figs. S4 and S5), respectively, which ensured a solid foundation for its applications. Molecules 9(10H)-acridone and diphenylsulfone were then ultilized as **guest** and **host** to construct the host-guest doping system, respectively. After mixing the solution of **guest** and **host** at certain mass ratio, the solvent was then removed by rotary evaporation. Finally, the as-prepared doping powders **DS-a** with the mass ratios of 1:100, 1:1000 and 1:10000 were obtained. A control group of the doping samples was also prepared with mass ratios of 1:1000 upon thermal melting-cooling (**DS-m**) or mechanical grinding (**DS-g**) treatment.

## Photophysical properties of host and guest

As shown in Fig. 2a, the UV–vis spectra of **guest** in 2-MeTHF exhibited two vibration absorption bands around 250–310 and 320–400 nm, which are attributed to locally excited states. Powder of **guest** showed similar absorption bands, but the range is broader and extends to 480 nm. The 77 K photoluminescence spectra of **guest** solution had two emissions located at 390–467 and 467–650 nm, which belong to fluorescence and phosphorescence, respectively (Fig. S6a). Interestingly, the fluorescence of **guest** gradually weakened after the addition of poor solvent n-hexane (Fig. S7), indicating the effect of aggregation caused quenching (ACQ), which could also be confirmed by the comparison of fluorescence quantum yields of its THF solution (16.6%) and powder (0.9%). Though the emission of **guest** powder was weak at room temperature, its specific emission could be observed clearly at low temperature. Compared to its solution, **guest** powder exhibited red-shifted fluorescence around 426–542 nm and phosphorescence around 530-720 nm at 77 K (Fig. S6a). Due to definite ACQ characteristics, the **guest** should be competent for a candidate to act as an energy acceptor when dispersed in the host matrix.

**Host** has shorter absorption and fluorescence emission wavelength because of weak conjugate structure. The absorption and emission spectra of **host** in both unimolecular state (solution) and aggregated state (solid) were then investigated. As shown in Fig. 2b, unimolecular **host** displays a vibrational absorption band between 240 and 285 nm, and the tail absorption is further extended to 350 nm once the molecules aggregate. In addition, a wide and weak absorption band occurs between 350 and 450 nm, which may derive from the absorption of intermolecular charge transfer. At 77 K, unimolecular **host** has the fluorescence band from 270 to 335 nm and the phosphorescence band from 348 to 520 nm (Fig. S6b). The corresponding phosphorescence lifetime exhibited double exponential decay, which implied the

existence of two phosphorescence emission processes from double triplet excited states (Fig. S8 and Table S2).

Surprisingly, the powder of **host** has two types of afterglows at low temperatures (Fig. 2c). The time-dependent spectra, emission decay curves and dual-exponential emission in solution of **host** combine to show that the shorter wavelength emission at 425 nm originates from the higher triplet state $T_2$ (752.1 ms at 77 K and 149.1 ms at 298 K, Figs. S9a and S9b), while the longer wavelength emission at 480 nm from the low-lying $T_1$ state (2360.7 ms at 77 K and 326.3 ms at 298 K, Figs. S9c and S9d), which suggests an anti-Kasha behavior. This property is similar to that of previous reported molecules[51,81,82], which also generated phosphorescence from $T_2$ and $T_1$ in the aggregated state. However, the phosphorescence quantum yield of **host** powder at room temperature is only 0.48% (Table S1), indicating that most of the excitons are lost through non-radiative process. The pathway for the non-radiative decay is likely through the strong ISC between singlet and triplet state; the triplet excitons are vulnerable to thermal motion, collision, and they are easily deactivated through quenching by air. Given the dual phosphorescence property of **host**, it is possible to construct a doping system by triplet-singlet ET to improve the exciton utilization and solve aggregation quenching. Stimuli-responsive luminescence of doping system can be also achieved through regulating host-guest packings.

## Photophysical properties of the host-guest doping system

The photophysical properties of **DS-a 1:1000** were then studied in detail (Figs. 3 and S10b). According to its excitation-afterglow profiles (Fig. S11), the excitation wavelengths of 295 nm and 374 nm were chosen to ensure the integrity of the absorption spectrum and PL spectrum of **host**. As shown in Fig. 3a, the delayed spectrum of **DS-a 1:1000** upon 374 nm excitation shows multiple emission peaks, which is likely a combination of multiple emissions from different excited states. The afterglow emission range and wavelength of **DS-a** 1:1000 combined together overlapped well with the superposition of prompt fluorescence of **guest** in the unimolecular state (solution) and aggregated state (powder) at 77 K (nor delayed emission of **guest** in unimolecular state), indicating that they are correlated. To our surprise, an unusual phenomenon occurred when **DS-a 1:1000** was exposed to 295 nm UV light. As shown in Fig. S10b, **DS-a 1:1000** emits obvious prompt PL between 300 and 390 nm and long wavelength delayed emission between 390 and 750 nm after 295 nm excitation. The emission around 300-390 nm is prompt fluorescence because its lifetime is only 0.2 ns. The 295 nm-excited delayed PL spectrum of **DS-a 1:1000** also shows multiple emission peaks, which is similar to that of 374 nm-excited, except that a new generated red-shifted emission peak appears at 490 nm. Depending on similar wavelengths and peak

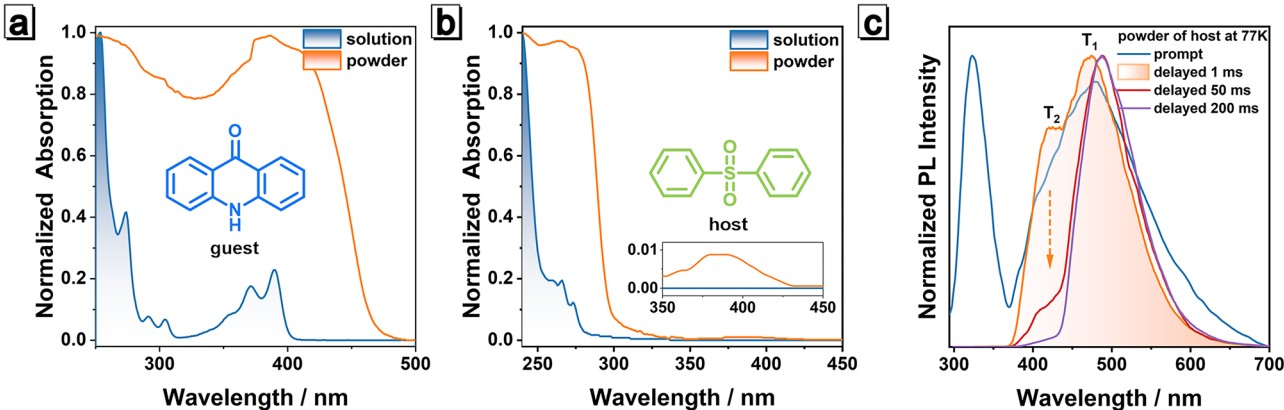

**Fig. 2 | Photophysical properties of guest and host.** Normalized absorption spectra of the 2-MeTHF solution (10 µM) and powder of **a** guest and **b** host at room temperature. **c** Normalized steady-state and delayed photoluminescence (PL) spectra of **host** powder at 77 K.

shapes and the delayed PL spectrum upon 295 nm excitation matches well with the prompt fluorescence of unimolecular/aggregated **guest** and $T_1$ phosphorescence of aggregated **host** (Fig. 3b). Though similar phenomenon has been rarely reported[67,68,72], important clues could be confirmed by the decay curves and the corresponding emission lifetimes (Figs. S12, S9b and S9d). The newborn emission at 490 nm possesses a long lifetime of ~380 ms upon the 295 nm excitation, which differs from ~135 ms short lifetime of the emissions upon 374 nm excitation, revealing the existence of another afterglow ($T_1$ of the aggregated **host**, Fig. S9d). As a result, the delayed PL spectrum shows abnormal red-shift emission with the shorter excitation wavelength, and their color difference can be distinguished even by naked eyes, indicating impressive Ex-De afterglow characteristic (Fig. 1b).

To further support our surmise on the composition of the delayed PL spectrum, more experiments were carried out. Upon 374 nm excitation, normalized delayed spectra show that the proportion of emissions at 400 and 424 nm significantly decreases with the increase of the doping ratio of **DS-a**, while the 472 nm emission increases (Fig. 3c and S10). This trend makes sense as the aggregation of **guest** is enhanced with the increase of doping ratio and the combined sources from aggregated and unimolecular **guest** also become more significant. Furthermore, the peak shape of delayed PL spectra of **DS-a 1:1000** after 374 nm excitation shows no significant variation with different delayed times (Fig. S13) and the lifetimes at different wavelengths are essentially the same (Table S3), confirming that the

afterglow should only come from the same photophysical process (ET from $T_2$ in **host** to $S_1$ in **guest**, discussed below) but different excited states. These experiments prove that the afterglow of **DS-a** upon 374 nm excitation can be deemed as delayed fluorescence derived from $S_1$ of both aggregated and unimolecular **guest**.

To figure out what causes the difference between the emissions upon the excitation of 295 nm and 374 nm, we tried to scrutinize related clues. On the basis of the absorption characteristics of **host** powder, we deduced that the whole process from excitation to emission is accompanied by ET. In Fig. 4a, the prompt emission of **host** powder partially overlaps with the absorption of **guest** solution and **guest** powder (shaded area). In contrast, the 323 nm emission of **host** powder overlaps less with the absorption of the **guest** solution, thus the fluorescence emission of aggregated **host** retains in doping system. The 360–400 nm absorption of **guest** solution and 360–460 nm absorption of **guest** powder are exactly close to $T_2$ of **host**. The sufficient overlap indicates that ET from $T_2$ of **host** to **guest** is practicable. To the best of our knowledge, ET from higher energy levels is rarely reported in previous literature. Moreover, the absorption of **guest** exactly covers $T_2$ and involves less with $T_1$, which will effectively reduce the influence from $T_1$ phosphorescence of **host**. The involvement of $T_2$ in ET is also demonstrated by the similar lifetimes of 400–460 nm emission of **DS-a** and that of $T_2$ emission of **host** powder. As shown in Fig. S14 and Table S3, lifetime of $T_2$ (~150 ms) is very close to that of 400-460 nm emission of **DS-a** with different doping ratios, which

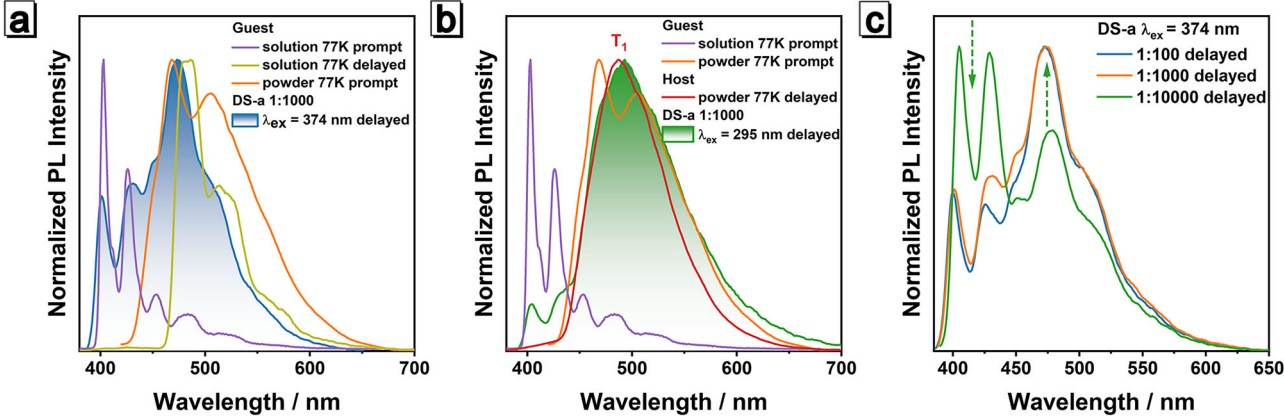

**Fig. 3 | Comparison of PL spectra of guest, host and DS-a.** Comparison of PL spectra of **DS-a 1:1000** under **a** 374 nm and **b** 295 nm excitation with those of **host** and **guest** in different environment. **c** Normalized delayed PL spectra of **DS-a 1:100**, **DS-a 1:1000** and **DS-a 1:10000** after 374 nm excitation.

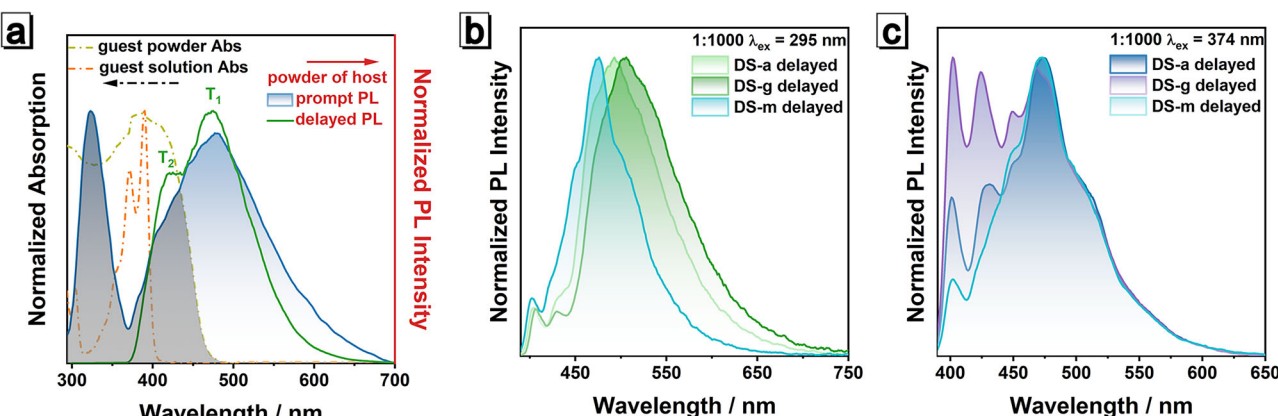

**Fig. 4 | Photophysical properties of guest, host and different host-guest doping systems. a** Normalized absorption of the powder and the solution of **guest**, and normalized steady-state and delayed PL spectra of **host** powder; dash area is the overlap of prompt emission of **host** powder and absorption of **guest** powder and **guest** solution. Normalized delayed PL spectra of **DS-a**, **DS-m** and **DS-g** with a doping ratio of 1:1000 under **b** 295 nm and **c** 374 nm excitation.

strongly proves that the energy is transferred from $T_2$ of **host** to **guest**. The prompt PL spectra of **DS-a** with different doping ratios under excitation of 295 nm indicates that the afterglow proportion increased significantly with the increase of doping ratio (Fig. S15), especially ~460 nm emission, suggesting that the increase of doping ratio not only improve the ET efficiency, but also increase the emission proportion from $S_1$ of aggregated **guest**. Ample evidence indicates that the ET process does exist consistently and plays a key role in this host-guest system.

Although we confirm that ET of $T_2$ leads to the long-lifetime delayed fluorescence of **guest** in **DS-a**, it is still difficult to explain why the 295 nm excitation can selectively induce $T_1$ phosphorescence of **host**. Based on current research, we propose a hypothesis that the generation of $T_1$ phosphorescence is related to the degree of light absorption by **host**. That means that the process depends on the number of the excitons produced. When the excitation wavelength is shorter than 295 nm, aggregated **host** absorbs enough energy and singlet excitons can be fully generated. These singlet excitons return to the ground state by fluorescence or non-radiative transition and reach $T_2$ state by ISC process; in this process, sufficient $T_2$ excitons not only cause significant fluorescence of **guest** through ET, but also transfer to $T_1$ through IC process accompanying with $T_1$ phosphorescence of **host**. When **DS-a** is excited with the wavelength longer than 350 nm, less excitons are produced due to weak absorption of aggregated **host**, resulting in insufficient $T_2$ excitons. Therefore, $T_2$ excitons preferentially transfer energy to **guest** and few excitons return to $T_1$ state, probably resulting in phosphorescence silence of $T_1$. To prove this hypothesis, we need to design new authentication experiments. It is well known that ET requires not only the overlap of absorption range of energy acceptor and emission range of energy donor, but also short intermolecular spacing. To improve the utilization efficiency of triplet excitons and reduce the intermolecular distance, mixed **host** and **guest** were thermal-melted and then cooled to prepare thermal crystallization sample (**DS-m 1:1000**). Thermal crystallization would provide compact packing and rigid environment to prevent oxygen and moisture from quenching the triplet excitons. As a result, **DS-m 1:1000** exhibit more significant afterglow compared to relative loose **DS-a 1:1000**, as verified by the increase of afterglow proportions (Fig. S16). **DS-m 1:1000** does not exhibit Ex-De afterglow (Figs. S17 and 1b), but its delayed spectra are composed of the delayed fluorescence from $S_1$ from the aggregated and unimolecular guest. Their delayed

fluorescence is nearly the same irrespective of 295 nm or 374 nm excitation (Fig. 4b, c), reflecting that ET efficiency is largely improved and IC process from $T_2$ to $T_1$ is almost suppressed. On the contrary, when **host** and **guest** were mixed and mechanically ground to obtain **DS-g 1:1000**, the proportion of >480 nm emission after 295 nm excitation increases (Figs. 4b and S18), indicating the ET process is weakened owing to long intermolecular distances and weak intermolecular interactions. The emission intensity ratio of 472 nm to 400 nm of **DS-g 1:1000** after 374 nm excitation is lower than that of **DS-a 1:1000** or **DS-m 1:1000** (Fig. 4c), which should relate to the low utilization efficiency of aggregated **guest** under loose packing environment. The loose molecular packing of **DS-g 1:1000** is also not conducive to the utilization of triplet excitons, resulting in the significantly low quantum yield of **DS-g 1:1000** after 374 nm excitation (Table S1). Powder XRD patterns of different doping samples and single crystal of **host** were then measured and showed in Fig. S19. All the doping samples have well-ordered crystalline structure with multiple similar diffraction peaks from $2\theta = 5°-25°$ with those of the single crystal of **host**, suggesting that the crystalline morphology in **host** is basically maintained after the doping of **guest**. Apart from relative intensity of characteristic peaks, the diffraction patterns of all doping samples resemble each other, which probably derive from subtle discrepancies of stacking density, dispersion of **guest**, etc. These subtle discrepancies impact the energy transfer efficiency and ultimately endow the doping samples doping method-dependent afterglow characteristic. The above evidences prove our hypothesis and fully explain the mechanism of Ex-De afterglow, paving a way for material design and theoretical improvement.

### Mechanism of the Ex-De afterglow of the host-guest system

According to the above experimental evidences, we proposed anti-Kasha triplet energy transfer principles. As shown in Fig. 5a, when the host-guest system is excited at 295 nm, **host** absorbs enough energy to generate abundant excitons. Some excitons are consumed in the form of prompt fluorescence, but the residual reaches $T_2$ through the ISC process. $T_2$ excitons can undergo ET process from $T_2$ of **host** to $S_1$ of **guest** and the arrival $S_1$ excitons are responsible for the delayed fluorescence. The other $T_2$ excitons transform into $T_1$ excitons through IC process, and subsequently produce $T_1$ phosphorescence of **host**. However, not enough excitons are produced due to weak absorption of **host** under 374 nm excitation, resulting in the low production of $T_2$

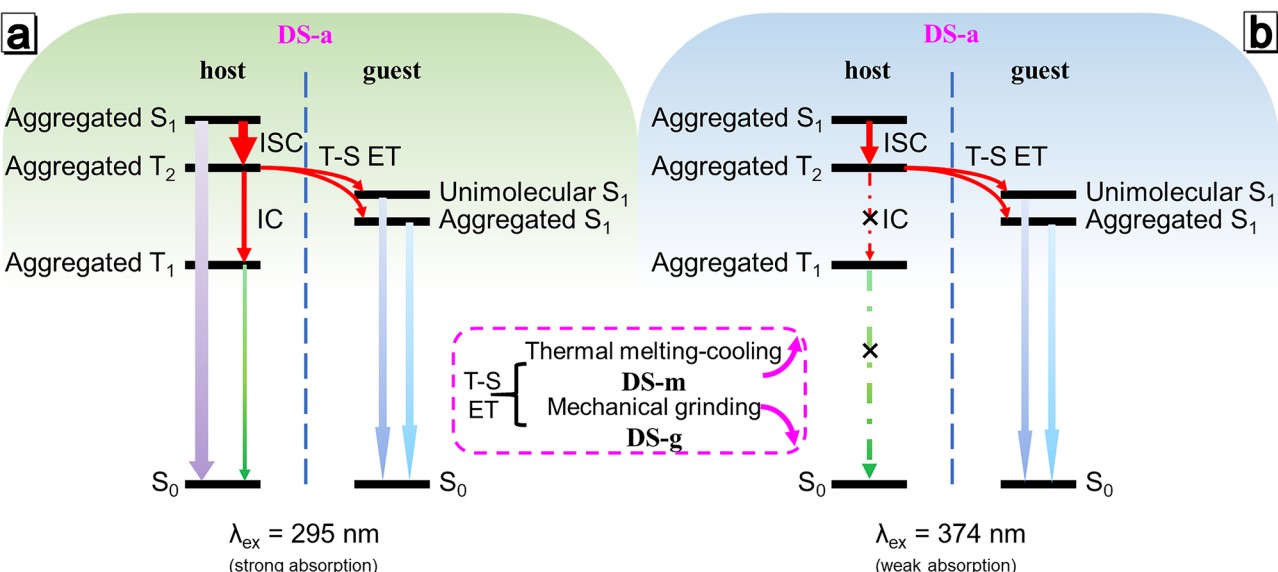

**Fig. 5 | A proposed mechanism of the afterglow behavior in the host-guest doping system.** A proposed mechanism of photophysical process of doping systems under **a** 295 nm and **b** 374 nm excitation.

excitons. Since $S_1$ of **guest** is closer to $T_2$ of host, the triplet-singlet ET efficiency should be higher than the $T_2$-$T_1$ IC efficiency. Preferential ET leads to consumption of most of the $T_2$ excitons, so only the delayed fluorescence of **guest** is observed after 374 nm excitation (Fig. 5b). When the doping system was prepared by thermal melting-cooling (**DS-m**) or grinding (**DS-g**), intermolecular spacing and environment should be diverse from those in **DS-a**, which led to an increase or decrease of ET efficiency (Figs. S20 and S21), respectively. Thus, the delayed spectra of this host-guest doping system can be tuned through different stimuli: **DS-m** shows single afterglow, while **DS-g** or **DS-a** exhibits Ex-De afterglow after 295/374 nm excitation (Fig. 1). Due to the increase of exciton utilization after doping, the quantum yield of the doping systems has a significant performance improvement compared to that of the pure **host** or **guest** (Table S1), achieving synergetic reinforcing effect.

Benzophenone is a commonly used host and has a similar structure with diphenylsulfone. For comparison, a new host-guest doping system **DS-BP** based on benzophenone was constructed through liquid mixing (Fig. S22a). However, no obvious afterglow of **DS-BP** could be observed by naked eyes. The absorption spectrum of aggregated benzophenone has no separated absorption band compared with that of diphenylsulfone. The similarity between the prompt and delayed PL of benzophenone powder at room temperature or 77 K, as well as the decay tendencies demonstrate that its emission essentially originates from phosphorescence (Figs. S22b and S22c). The same lifetimes at different wavelengths prove that the delayed PL comes from a single emission state of $T_1$. As shown in Figs. S22d and S22e, the delayed PL of **DS-BP** around 600 nm can be attributed to $T_1$ of **guest**, while the band between 400 and 550 nm is derived from $T_1$ of benzophenone, which can be proved by their different lifetimes and spectral contours. Unlike the phenomena in **DS-a/DS-m/DS-g**, the delayed PL of **DS-BP** has no characteristic of prompt fluorescence of unimolecular **guest**, which probably point to the mismatch between $T_1$ of aggregated benzophenone and absorption of unimolecular **guest** (Fig. S22f). These results further verify the rationality of anti-Kasha triplet energy transfer and Ex-De afterglow behavior in **DS-a**.

## Applications of host-guest doping system

Since the **host** and **guest** can be mixed via facile grinding, solution evaporation or heating, the doping systems is of great potential to serve as security label, anti-counterfeiting/display ink or thermal printing paper (Fig. 6). Based on the force-triggered RTP effect of the doping system, prototype of security label was prepared. Powders of **guest** and **host** (1:100) were manually shaken to ensure a certain degree of uniformity. Then the mixture was cast on the glass substrate and sealed with scotch tape in order to avoid material leakage. The as-prepared label could not emit RTP after 365 nm excitation. When scraping on the scotch tape with a metal spatula or something hard, the blue RTP of the ground region was switched on and formed strong contrast (Fig. 6a).

By mixing of the **host** and **guest** in liquid form, anti-counterfeiting/display ink could be designed (Fig. 6b). Filter paper was dip-coated with ethanol solution of **guest** (0.5 mg/mL) and naturally dried. The whole guest-loaded paper was dark after turning off the UV light by virtue of the absence of RTP activity. When ethanol solution of **host** (20 mg/mL) was drawn on this filter paper with a Chinese brush to allow to evaporate, **host** and **guest** mixed together to form **DS-a**. After turning off 365 and 254 nm excitation light, the letters with bluish and greenish RTP emission could be observed clearly, respectively. This dynamic RTP phenomenon would promote the effect of anti-counterfeiting. Following the same principle, **host** and **guest** could also be pre-mixed in solution to prepare display inks and used for inkjet printing.

Through thermal mixing of **host** and **guest**, a thermal printing paper was designed as followed. Weighing paper was cut into a proper size and then soaked in ethanol solution of **guest** (0.5 mg/mL) to prepare the **guest**-loaded paper. After the solvent evaporated, fine powder of host was gently smeared on one side of the **guest**-loaded paper. When the thermal paper passed through the print head of thermal printing machine, **guest** and **host** would partially blend together to form **DS-m** under the action of heating due to the low melting point of **host**. As shown in Fig. 6c, two sky-blue emissive Chinese characters (meaning 'Shenzhen') could be observed by naked eyes upon 365 or 254 nm excitation, corresponding to the formation of **DS-m**. Different kinds of letters or characters could also be printed according to specific requirements. Utilizing this doping system, multiple anti-counterfeiting/display applications could be successfully achieved with the unique stimuli-responsive RTP characteristic.

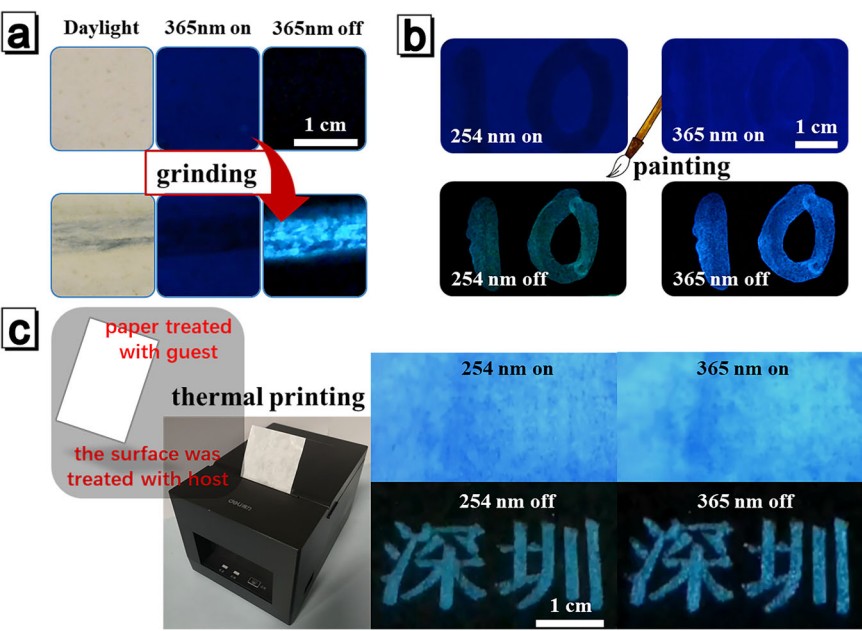

**Fig. 6 | Stimuli-responsive afterglow property of host-guest doping system and corresponding applications. a** Prototype of security label. **b** Prototype of anti-counterfeiting/display ink. **c** Application of thermal printing paper.

## Discussion

In summary, we have successfully developed an organic host-guest doping system, in which 9(10H)-acridone acted as **guest** and diphenylsulfone as **host** to turn-on afterglow under 365 nm excitation via facile grinding, solution evaporation or heating. This doping system possessed several characteristic features: (1) Ex-De afterglow, which is a very useful property for non-invasive tuning the emission of materials and rarely observed among two-component small molecule doping systems. (2) Anti-Kasha's rule, ET process from higher excited triplet state $T_2$ of **host** to the lowest excited singlet state $S_1$ of the aggregated/unimolecular **guest** plays a crucial role in the Ex-De afterglow behaviors, which is even rarer. (3) Multiple processing methods, the Ex-De afterglow and RTP turn-on phenomenon could be achieved through grinding, writing, painting, or thermal printing. Thanks to these features, we successfully realized applications in multiple information anti-counterfeiting and display. This work opened up a way for the preparation of two-component small molecule Ex-De RTP materials.

## Methods

### Materials

9(10H)-Acridone (98%), diphenyl sulfone (99%), and benzophenone (99%) were purchased from Energy Chemical Company. Solvents (AR) were purchased from Xilong Chemical Co., Ltd.

### Instruments and methods

All photophysical measurements, including steady or delay spectra, decay curve, and quantum yield, were carried out on Edinburgh Instruments FLS1000 with different accessories. [1]H and [13]C NMR spectra were recorded on a VNMRS 400 NMR spectrometer (Varian, USA). Sample heating was performed on a microcomputer temp-controlled heating board (JF-966A, JFTOOLS, China). TGA analysis was performed on a NETZSCH SA409PC thermogravimeter. Photos were taken by Nikon D7100 or HUAWEI P20. High-performance liquid chromatogram (HPLC) curves were recorded using an Agilent 1260 Infinity II(QA&QC-HPLC-19) by DAICEL CHIRAL TECHNOLOGIES(CHINA) CO., LTD. Thermal printing was performed on a thermal printing machine (DL-581PS, Deli, China).

**Purification of host.** The high purity **host** was obtained after recrystallizing from its ethanol and then acetone/$H_2O$ solution.

**Purification of guest.** The high purity **guest** was obtained after recrystallizing from its acetic acid and then methanol solution.

**Purification of benzophenone.** Benzophenone was obtained after recrystallizing from its petroleum ether solution.

## Data availability

All relevant data are included in this Article and its Supplementary Information files. Data is available from the authors upon request. Source data are provided in this paper.

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

## Acknowledgements

This work was supported by the National Nature Science Foundation of China (21905177 and 21975061) (G.X. and Z.H.) and the Natural Science Foundation of Guangdong Province (2019KZDXM008 and 2021A1515010192) (B.S.L.) and Shenzhen Fundamental Research Program (JCYJ20210324094607021 and JCYJ20190806142403535) (B.S.L. and Z.H.).

## Author contributions

Conceptualization: B.S.L., G.H., and Z.H. Methodology: G.H., Z.H., and B.S.L. Investigation: W.H. and W.X. Visualization: W.X., W.H., and J.L. Funding acquisition: G.H., B.S.L., and Z.H. Project administration: G.H. and Z.H. Supervision: B.S.L. and B.Z.T. Writing (original draft): W.H. and W.X. Writing (review and editing): G.H., B.S.L., Z.H., and B.Z.T.

## Competing interests

The authors declare no competing interests.
