## [Peer Review File · Nature Communications]

Anti-Kasha Triplet Energy Transfer and Excitation Wavelength Dependent Persistent Luminescence from Host-Guest Doping SystemsREVIEWER COMMENTS

Reviewer #1 (Remarks to the Author):

Anti-Kasha's emission in organic luminogens has attracted many attentions since its discovery. Li et al. reported an excitation-wavelength dependent afterglow doping system based on anti-Kasha rule of energy transfer from triplet T2 of host to singlet S1 of guest, which is less-developed and exploited. The applications of the doping system's multiple stimuli-responsive properties in information anti-counterfeiting and display were also demonstrated. In the manuscript, the origins of each emission were fully verified, and excitation-emission processes were explained in detail. I consider this work is worth publishing in Nature Communications after some issues and improvements are addressed, as described below.

- 1) Figure 5 showed the mechanism of the afterglow behavior of the host-guest doping system. However, it is unclear why the efficiency of energy transfer is higher than the efficiency of internal conversion within the triplet state?
- 2) In Figure 2b, the absorption spectrum of Host shows a wide and weak absorption band between 350 and 450 nm, which may derive from the absorption of intermolecular charge transfer. So whether it is reasonable that the energy transfer starts from the aggregated S1 of Host in Figure 5b?
- 3) Why no phosphorescence of the Guest could be observed in the doping system?
- 4) Have the authors tried other commonly used host molecules to substitute diphenylsulfone? Whether the new systems have RTP or excitation-wavelength dependent afterglow?
- 5) There are several spelling errors, such as effeicent, porcess, etc. The authors should carefully check and correct them.

Reviewer #2 (Remarks to the Author):

In this paper, Li and Tang et al. reported a stimulus-responsive host-guest system which achieved morphology-dependent and excitation-dependent afterglow with improved luminescent efficiency. Surprisingly, an unusual anti-Kasha triplet energy transfer was realized by energy level matching between guest and host, and the results endowed multiple photophysical process with potential application prospect. In my opinion, the manuscript is suitable for publication in Nature Communications for the potential research significance. Therefore, the manuscript was suggested to be accepted after minor revision.

Several comments:

1. The authors have discussed the emission behavior of doping systems under different morphologies, but the details of morphologies were not provided. I considered the PXRD of three different morphologies should be characterized and compared with that of the single crystal of the host. The relevant data needed to be supplemented to enrich the manuscript's findings.

2. Two excitation wavelengths were selected as the working condition of light source. Whether there exist other excitation wavelengths that cause changes in the afterglow. It might be worthwhile for the authors to check the possibility of other excitation wavelengths.
3. If the afterglow excited at 374nm only comes from the S1 of the guest, why the prompt PL of the doping system is not coincidence with its delayed PL?
4. The details of absorption between 350 and 450 nm in Fig. 2b needed to be enlarged.
5. Page numbers of references should be checked, several of them was missing.

Responses to Reviewer #1

Anti-Kasha's emission in organic luminogens has attracted many attentions since its discovery. Li et al. reported an excitation-wavelength dependent afterglow doping system based on anti-Kasha rule of energy transfer from triplet T₂ of host to singlet S₁ of guest, which is less-developed and exploited. The applications of the doping system's multiple stimuli-responsive properties in information anti-counterfeiting and display were also demonstrated. In the manuscript, the origins of each emission were fully verified, and excitation-emission processes were explained in detail. I consider this work is worth publishing in Nature Communications after some issues and improvements are addressed, as described below.

Response: Thank you very much. The manuscript had been carefully revised according to your important suggestions. For your convenience, the main revisions are marked with yellow.

1) Figure 5 showed the mechanism of the afterglow behavior of the host-guest doping system. However, it is unclear why the efficiency of energy transfer is higher than the efficiency of internal conversion within the triplet state?

Response: Thanks for your professional question. The occurrence of anti-Kasha's emission is the result of the competition between the radiation rate (k_{rad}) and the internal-conversion rate (k_{IC}), as long as k_{rad} is high enough or k_{IC} is low enough. In general, k_{IC} is easily regulated due to differences in molecular structure. Thus, it is possible to obtain anti-Kasha's behavior through reduced k_{IC} . Slow internal conversion may result from either a poor Franck-Condon factor (when the energy gap between T₂ and T₁ is large), or a poor electronic factor between T₂ and T₁ (DOI: 10.1021/acs.chemrev.7b00110). The doping system reported in this manuscript is clearly ascribed to former reason, that is, the generation of anti-Kasha's emission from T₂ is due to the large energy difference between T₂ and T₁. From delayed PL at 77 K, the energy gap between T₂ and T₁ ($\Delta E_{\text{T}_2\text{-T}_1}$) in **host** is 0.46 eV, which is competent for suppressing internal conversion. Similarly, energy transfer also depends on the proximity of energy levels. When energy level of **guest** is close enough to that of **host**, accompanied by the reduction of the distance between **guest** and **host**, the energy transfer can be greatly improved. Therefore, through the inhibition of internal conversion and the enhancement of energy transfer, the efficiency of energy transfer is higher than the efficiency of internal conversion within the triplet state may occur.

2) In Figure 2b, the absorption spectrum of Host shows a wide and weak absorption

band between 350 and 450 nm, which may derive from the absorption of intermolecular charge transfer. So whether it is reasonable that the energy transfer starts from the aggregated S₁ of Host in Figure 5b?

Response: By comparing the absorption spectra of unimolecular **host** and aggregated **host** in Figure 2b, it can be found that the band between 350 and 450 nm is indeed the intermolecular-induced absorption caused by aggregation. The wide and weak absorption band exhibits a vibration-free and red-shift dynamics, satisfying the condition of intermolecular charge transfer. Therefore, it can be inferred that this absorption is attributed to S₁ of J-type aggregation of **host** with intermolecular charge transfer characteristics. Moreover, with this low-level aggregated S₁, the energy gap (ΔE_{ST}) between the single and triplet state is reduced, which may be conducive to intersystem crossing and promote the following triplet energy transfer.

3) Why no phosphorescence of the Guest could be observed in the doping system?

Response: From prompt PL and delayed PL of **guest** at 77 K in Supplementary Figure 6, it can be observed that **guest** possesses weak intersystem crossing, which is not conducive to the generation of phosphorescence. Moreover, the energy gap between the triplet state in **guest** and the singlet state in **host** is significant, making the doping system difficult to undergo singlet-triplet energy transfer. These results probably cause the phosphorescence of **guest** is too weak to be observed in the doping system.

4) Have the authors tried other commonly used host molecules to substitute diphenylsulfone? Whether the new systems have RTP or excitation-wavelength dependent afterglow?

Response: Thanks for your professional revision and valuable suggestion. We chose benzophenone as host to construct a new host-guest doping system **DS-BP** for its similar structure with diphenylsulfone. The new doping system has very weak RTP so the discussion on excitation-wavelength dependent afterglow becomes unnecessary.

Relevant discussions have been added in the manuscript. “Benzophenone is a commonly used host and has a similar structure with diphenylsulfone. For comparison, a new host-guest doping system **DS-BP** based on benzophenone was constructed through liquid mixing (Fig. S22a). However, no obvious afterglow of **DS-BP** could be observed by naked eyes. The absorption spectrum of aggregated benzophenone has no separated absorption band compared with that of diphenylsulfone. The similarity between the prompt and delayed PL of benzophenone powder at room temperature or

77 K, as well as the decay tendencies demonstrate that its emission essentially originates from phosphorescence (Fig. S22b and S22c). The same lifetimes at different wavelengths prove that the delayed PL comes from a single emission state of T_1 . As shown in Fig. S22d and S22e, the delayed PL of **DS-BP** around 600 nm can be attributed to T_1 of **guest**, while the band between 400 and 550 nm is derived from T_1 of benzophenone, which can be proved by their different lifetimes and spectral contours. Unlike the phenomena in **DS-a/DS-m/DS-g**, the delayed PL of **DS-BP** has no characteristic of prompt fluorescence of unimolecular **guest**, which probably point to the mismatch between T_1 of aggregated benzophenone and absorption of unimolecular **guest** (Fig. S22f). These results further verify the rationality of anti-Kasha triplet energy transfer and Ex-De afterglow behavior in **DS-a**.”

Supplementary Figure 22 (a) The new doping system **DS-BP** based on benzophenone host. (b) Normalized absorption spectrum, prompt and delayed PL of benzophenone powder. (c) Decay curves of delayed emissions of benzophenone powder at room temperature. (d) Comparison of prompt and delayed PL spectra of **DS-BP** with those of **guest** and benzophenone powder. (e) Decay curves of delayed emissions of **DS-BP** at room temperature. (f) Normalized absorption of the powder and the solution of **guest**, and normalized delayed PL spectra of benzophenone powder.

5) There are several spelling errors, such as *effeicient*, *porcess*, etc. The authors should carefully check and correct them.

Response: The spelling errors in the Introduction had been revised. Thank you very much.

Responses to Reviewer #2

In this paper, Li and Tang et al. reported a stimulus-responsive host-guest system which achieved morphology-dependent and excitation-dependent afterglow with improved luminescent efficiency. Surprisingly, an unusual anti-Kasha triplet energy transfer was realized by energy level matching between guest and host, and the results endowed multiple photophysical process with potential application prospect. In my opinion, the manuscript is suitable for publication in Nature Communications for the potential research significance. Therefore, the manuscript was suggested to be accepted after minor revision.

Response: Thank you very much. The manuscript had been carefully revised according to your important suggestions. For your convenience, the main revisions are marked with yellow.

Several comments:

1. The authors have discussed the emission behavior of doping systems under different morphologies, but the details of morphologies were not provided. I considered the PXRD of three different morphologies should be characterized and compared with that of the single crystal of the host. The relevant data needed to be supplemented to enrich the manuscript's findings.

Response: The PXRD of three different morphologies and the single crystal of the **host** have been characterized and added as Supplementary Figure 19.

Relevant discussions have been added into the manuscript. “Powder XRD patterns of different doping samples and single crystal of **host** were then measured and showed in Fig. S19. All the doping samples have well-ordered crystalline structure with multiple similar diffraction peaks from $2\theta = 5^\circ$ - 25° with those of the single crystal of **host**, suggesting that the crystalline morphology in **host** is basically maintained after the doping of **guest**. Apart from the relative intensity of characteristic peaks, the diffraction patterns of all doping samples resemble to each other, which probably derive from subtle discrepancies of stacking density, dispersion of **guest**, etc. These subtle discrepancies impact the energy transfer efficiency and ultimately

endows the doping samples doping method-dependent afterglow characteristic.”

Supplementary Figure 19 XRD patterns of doping samples.

2. Two excitation wavelengths were selected as the working condition of light source. Whether there exist other excitation wavelengths that cause changes in the afterglow. It might be worthwhile for the authors to check the possibility of other excitation wavelengths.

Response: Thank you so much for your professional opinions. Regarding to this, the excitation-afterglow mapping of **DS-a 1:1000** under ambient condition was conducted and added as Supplementary Figure 11. According to its excitation-afterglow profiles, it can be concluded that there exist two different excitation bands: 280-300 nm and 360-400 nm. When excitation wavelength is shifted from 360-400 nm to 280-300 nm, additional emission appears at 500-550 nm, which is attributed to the phosphorescence of **host**. Moreover, the excitation of 280-300 nm produces stronger emission, in accordance with the strong absorption characteristics of **host**. Therefore, we have not found other potential excitation wavelengths that can cause changes in the afterglow from excitation-afterglow mapping.

Supplementary Figure 11 Excitation-afterglow mapping of **DS-a 1:1000** under ambient condition.

3. *If the afterglow excited at 374nm only comes from the S₁ of the guest, why the prompt PL of the doping system is not coincidence with its delayed PL?*

Response: Actually, S₁ excitons from **host** can still generate under the excitation of 374 nm, and these excitons can also be partially transferred to S₁ of **guest** via singlet-singlet energy transfer, which immediately produces transient fluorescence from the **guest** S₁. Thus, prompt PL of the doping system should contain both transient and delayed fluorescence from **guest**, leading to the difference between prompt PL and delayed PL.

4. *The details of absorption between 350 and 450 nm in Fig. 2b needed to be enlarged.*

Response: The detail of absorption between 350-450 nm in Fig. 2b has been enlarged.

5. *Page numbers of references should be checked, several of them was missing.*

Response: Page numbers of references 23, 38, 40, 56, 60 and 77 had been added. Thank you very much.

REVIEWERS' COMMENTS

Reviewer #1 (Remarks to the Author):

The authors addressed my concerns! Now I recommend that this manuscript can be accepted.

Reviewer #2 (Remarks to the Author):

The authors have revised the manuscript considerably. Therefore, We recommend the paper for publication in Nature Communications

Responses to Reviewer #1

The authors addressed my concerns! Now I recommend that this manuscript can be accepted.

Response: Thank you very much. The manuscript had been carefully revised according to author checklist. For your convenience, the main revisions are marked with yellow.

Responses to Reviewer #2

The authors have revised the manuscript considerably. Therefore, We recommend the paper for publication in Nature Communications

Response: Thank you very much. The manuscript had been carefully revised according to author checklist. For your convenience, the main revisions are marked with yellow.